# Grain Fe and Zn contents linked SSR markers based genetic diversity in rice

Qasim Raza[1]*, Awais Riaz[1], Hira Saher[1], Ayesha Bibi[2], Mohsin Ali Raza[3], Syed Sultan Ali[1], Muhammad Sabar[1]

1 Molecular Breeding Laboratory, Rice Research Institute, Kala Shah Kaku, Sheikhupura, Punjab, Pakistan,
2 Plant Pathology Laboratory, Rice Research Institute, Kala Shah Kaku, Sheikhupura, Punjab, Pakistan,
3 Rice Technology Laboratory, Rice Research Institute, Kala Shah Kaku, Sheikhupura, Punjab Pakistan

* qasimnazami@gmail.com

**Data Availability Statement:** All relevant data are within the manuscript and its Supporting Information files.

## Abstract

Rice is critical for sustainable food and nutritional security; however, nominal micronutrient quantities in grains aggravate malnutrition in rice-eating poor populations. In this study, we evaluated genetic diversity in grain iron (Fe) and zinc (Zn) contents using trait-linked simple sequence repeat (SSR) markers in the representative subset of a large collection of local and exotic rice germplasm. Results demonstrated that aromatic fine grain accessions contained relatively higher Fe and Zn contents in brown rice (BR) than coarse grain accessions and a strong positive correlation between both mineral elements. Genotyping with 24 trait-linked SSR markers identified 21 polymorphic markers, among which 17 demonstrated higher gene diversity and polymorphism information content (PIC) values, strongly indicating that markers used in current research were moderate to highly informative for evaluating the genetic diversity. Population structure, principal coordinate and phylogenetic analyses classified studied rice accessions into two fine grain specific and one fine and coarse grain admixture subpopulations. Single marker analysis recognized four ZnBR and single FeBR significant marker-trait associations (MTAs) contributing 15.41–39.72% in total observed phenotypic variance. Furthermore, high grain Fe and Zn contents linked marker alleles from significant MTAs were also identified. Collectively, these results indicate a wide genetic diversity exist in grain Fe and Zn contents of studied rice accessions and reveal perspective for marker-assisted biofortification breeding.

## Introduction

Rice is one of the most consumed food crops globally, especially in developing countries [1]. The world population is increasing at an exponential rate; however, current annual genetic gain in rice production is insufficient to meet future food requirements [2], demanding continuous efforts to breed high yielding and more nutritious rice cultivars. Thus, there is an urgent need to boost global rice production, along with nutritional quality, for sustainable food and nutritional security [3].

**Funding:** This research was financially supported by Punjab Agricultural Research Board Pakistan under a competitive research project grant number PARB 904 to Muhammad Sabar. The funders had no role in study design, data collection and analysis, decision to publish, or preparation of the manuscript.

**Competing interests:** The authors have declared that no competing interests exist.

Among essential micronutrients, iron (Fe) and zinc (Zn) are vital for maintaining a healthy lifestyle in both animals and plants [4, 5]. In humans, the 30% of daily estimated average requirement of Fe and Zn is 13–28 μg/g [6]. However, minimal quantities of both of these micronutrients are found in rice grains [7]. Furthermore, the practice of consuming polished rice grains in Asian populations aggravates malnutrition [8]. During recent past, proper attention has been given to improve rice grain Fe and Zn contents, as wide genetic variability for these essential micronutrients have been reported in natural rice germplasms [9–14]. Plant breeding based biofortification is the most cheaper and sustainable approach to improve grain micronutrient contents and eradicates malnutrition from rice-eating poor populations [15]. Enormous genetic potential of rice germplasms for grain Fe and Zn contents could be exploited through marker-assisted biofortification breeding for development of micronutrient dense rice cultivars.

Systematic understanding of the extent of genetic variability and genetic relationships among different genotypes are pre-requisites for effective plant breeding programs [16]. Evaluation of genetic diversity in local and exotic germplasms could be helpful in crop improvement and sustainable agriculture development. Generally, due to great diversity in climatic and edaphic factors, rice genotypes of South Asia possess some unique characteristics which are of great interest to the modern rice breeders. Some of these beneficial characteristics include aroma [17], relatively higher grain Fe and Zn contents [18], drought and heat tolerance [19], and high rice production [20, 21]. These unique traits could be exploited through conventional and molecular breeding approaches for the development of more nutritious and resilient high yielding cultivars.

Like other crop plants, several studies have been conducted in rice to evaluate genetic diversity in grain Fe & Zn contents, drought tolerant & susceptible genotypes and local & exotic germplasm using molecular markers [10, 13, 22, 23]. Among various molecular markers, simple sequence repeat (SSR) or microsatellite markers are most widely and preferentially employed for genetic diversity studies. SSRs are cost-effective, easy to score, rapid, reliable and require minimal quantities of DNA [24]. They can efficiently establish genetic relationships due to their extensive distribution across genome, relative polymorphic abundance and co-dominant nature [25, 26]. Although, recent studies have evaluated genetic diversity in diverse rice germplasm using several random and trait linked SSR markers [10, 13]; however, none of these have used grain Fe and Zn contents linked SSR markers. The present study was conducted to assess genetic diversity in the representative subset of a large local and exotic rice germplasm collection using grain Fe and Zn contents linked SSR markers. Furthermore, possible population structure, genotypic relationships and marker-trait associations were also investigated, which could facilitate the conservation and utilization of studied germplasm resources.

## Material and methods

### Plant materials

A total of 56 genetically diverse *Indica* and/ or *Japonica* derived rice accessions, representing a large collection of local and exotic rice germplasm, including landraces, farmer field improved cultivars, advance uniform lines in release pipeline, recombinant/near-isogenic/backcross inbred lines, fine grain, coarse grain, aromatic, non-aromatic and few sequenced genome accessions, originating from three major rice producing countries (Pakistan, China and India) were used in this study (S1 Table in S1 File). The seed of these accessions was obtained from Rice Research Institute, Kala Shah Kaku (RRI, KSK), Pakistan and International Rice Research Institute (IRRI), Philippines. All accessions were planted under natural field conditions at the

experimental area of RRI, KSK during two cropping seasons 2018 and 2019. The seed obtained from these experiments was used for estimation of Fe and Zn contents.

## Estimation of Fe and Zn contents

Brown rice samples were prepared for all accessions using paddy dehusker (Satake, Japan). Fe & Zn contents were estimated by following Estefan et al. [27] wet-digestion method. For this, one gram of sample was chemically digested with 10 ml of nitric acid and perchloric acid mixture (2:1). The mixture was heated in a cold-digestion block until turned colourless or white precipitation occurred. Afterwards, double-distilled deionized water was added for dissolving the crystals and diluting the extract to 50 ml. Finally, extract was filtered through Whatman No. 41/42 filter paper before feeding to atomic absorption spectrophotometer (200 Series AA, Agilent Technologies, USA) for quantification of mineral contents. The analysis was repeated thrice for each accession during two cropping seasons separately and mean values were expressed in parts per million (ppm = $\mu$g/g). Pearson correlation between grain Fe and Zn contents was computed using Statistix 8.1 and scatter plot was drawn by setting an intercept value of '0' with Microsoft Excel 2016.

## DNA isolation and SSR based genotyping

Genomic DNA was isolated by following modified cetyl tri-methyl ammonium bromide method [28] from leaf tissues collected during cropping season 2018. DNA quantity and quality were assessed using Nanodrop Spectrophotometer (ND 2000, Thermo Scientific, USA) and 0.8% agarose gel electrophoresis with 1X Tris-borate-EDTA buffer and ethidium bromide staining, respectively. Twenty-four SSR marker primer pairs were selected for genotyping based on their inherent association with grain Fe and Zn contents [29]. The detailed information of all SSR markers is given in S2 Table in S1 File. Series of polymerase chain reactions and 8% polyacrylamide gel electrophoresis were performed to detect SSR fingerprints by following our previously published protocol [30]. Molecular marker size of different alleles was determined using a 20 bp DNA ladder (Fermentas, USA).

## Genetic diversity analysis

Genetic diversity statistics generated by each SSR marker were estimated using PowerMarker v3.25 [31] and included number of alleles (Na), major allele frequency ($M_{AF}$), gene diversity ($G_D$), heterozygosity (Het.) polymorphism information content (PIC) and inbreeding coefficient (f). The possible population genetic structure was drawn using a Bayesian clustering method with STRUCTURE v2.3.6 program [32] after setting the burin-in period and MCMC repeats as 100,00 each. The optimal numbers of subpopulations were determined based on STRUCTURE HARVESTER results [33]. Principal coordinate analysis (PCoA) was performed using DARwin6.5 program [34]. A phylogenetic tree showing genetic relationships among sampled accessions was constructed with MEGA 7 [35] using pairwise dissimilarity matrix and unweighted pair group method with arithmetic mean (UPGMA) statistical method.

## Marker trait associations

Associations between markers and grain Fe and Zn contents were assessed by following quantitative trait loci (QTL) analysis with single factor ANOVA procedure in excel (https://passel2.unl.edu/view/media?view=animations). Different marker alleles were compared with Fe and Zn contents and box plots were generated using Microsoft Excel 2016.

## Results

### Fe and Zn contents in rice grains

To determine Fe and Zn contents, brown rice samples of 56 genetically diverse *Indica* and/ or *Japonica* derived rice accessions (S1 Table in S1 File) were processed through a wet-chemical digestion method and the extract was fed to atomic absorption spectrophotometer. Significant genetic variability existed among studied accessions for grain Fe and Zn contents (Fig 1A). Fe contents varied from 5.45 ppm (PK 10967) to 52.30 ppm (Basmati 370), with an average of 20.82 ppm. Similarly, Zn contents ranged from 7.39 ppm (UHL17081) to 76.35 ppm (Super Gold), with an average of 25.12 ppm. Fine grain accessions had relatively higher Fe (21.74 ppm) and Zn (26.08 ppm) contents than coarse grain accessions (Fe 18.88 ppm; Zn 23.09 ppm) (Fig 1B). Nearly, 18% and 23% of the total accessions contained > 30 ppm Fe and Zn densities respectively, and the majority of these had fine grains. Furthermore, significant positive correlation (r = 0.757, $p < 0.001$) was present between grain Fe and Zn contents (Fig 1C). These results indicate availability of significant genetic variation for further simultaneous improvement of both Fe and Zn contents, especially in fine grain rice accessions.

### Genetic diversity among rice accessions

DNA fingerprinting of 56 genetically diverse rice accessions was done using 24 trait linked SSR markers. The agarose gel pictures showing banding patterns of some rice accessions and summary statistics of genetic markers used in this study are presented in Fig 2 and Table 1, respectively. Out of 24 SSR markers, 21 (87.5%) were found to be polymorphic and these generated a total of 82 alleles. The number of alleles per locus ranged from 1 (RM 153, RM 1357 and RM 7414) to 9 (RM 335), with an average of 3.5 alleles per locus. Major allele frequency ($M_{AF}$) of all genetic markers ranged from 0.2679 (RM 335) to 1.0000 (RM 153, RM 1357 and

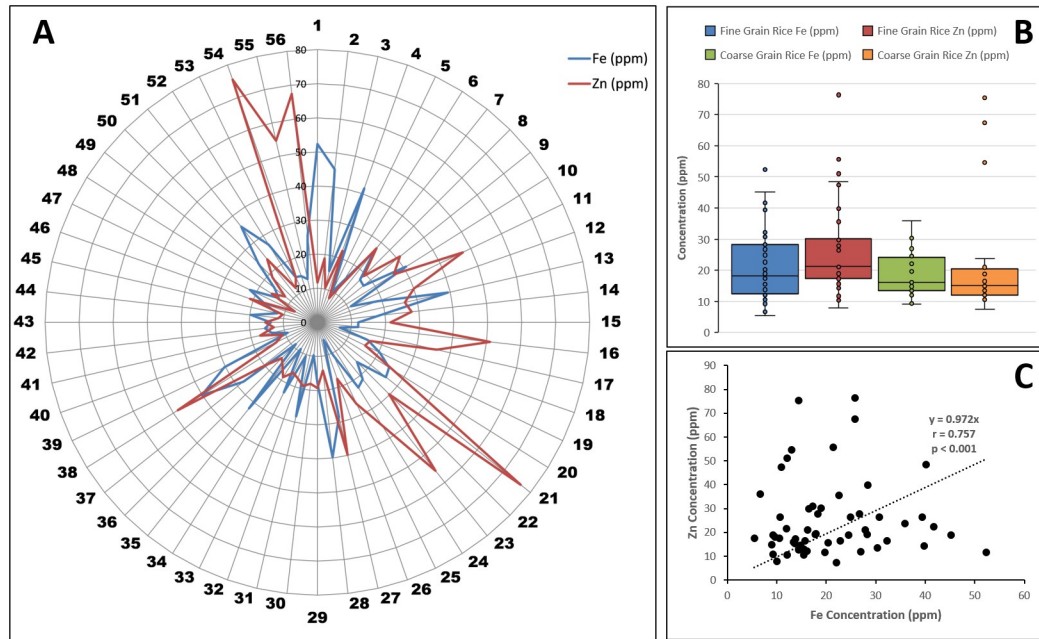

**Fig 1. Grain Fe and Zn contents and correlation in studied rice accessions.** (**A**) Grain Fe and Zn contents in studied rice accessions. Numbers outside the spider graph indicate serial numbers of the genotypes mentioned in S1 File. (**B**) Comparison between grain Fe and Zn contents in fine and coarse grain accessions. Each dot in box plots represents individual value. (**C**) Pearson correlation between Fe and Zn contents.

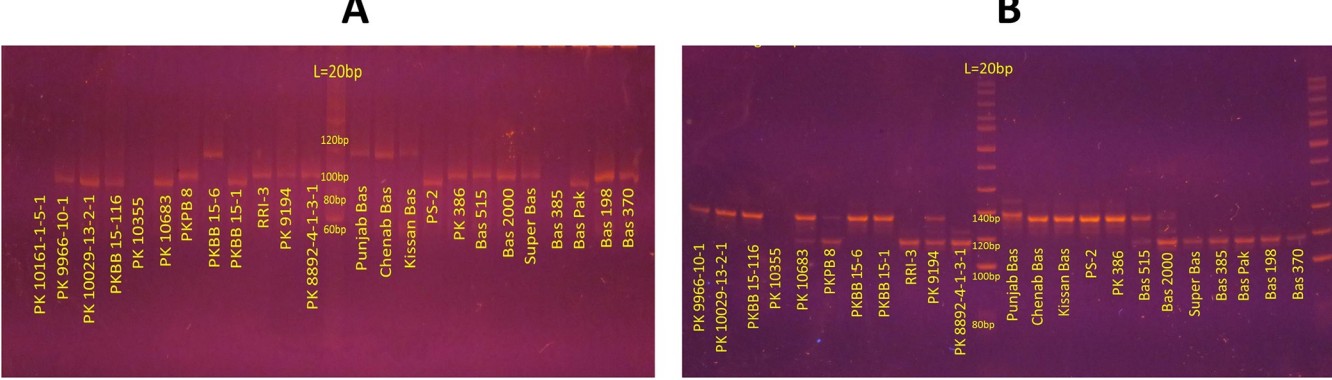

**Fig 2. DNA banding patterns in some rice accessions.** The 8% polyacrylamide gel electrophoresis pictures showing DNA banding patterns in some rice accessions generated with SSR marker (**A**) RM 335 and (**B**) RM 402. A 20 base pair ladder is presented in the middle of each figure.

RM 7414), with a mean of 0.5956. Gene diversity and observed heterozygosity varied from 0.1677–0.8182 and 0.0000–0.1964, respectively. Polymorphism information content (PIC) ranged from 0.1630 (RM 439) to 0.7960 (RM 335), with an average of 0.4663. Similarly, inbreeding coefficient (f) varied from 0.7052 (RM 430) to 1.0000 (RM 319, RM 541, RM 560 and RM 439). The summary of genetic diversity parameters of all SSR markers is presented in Table 1.

## Grouping of rice accessions by population structure, principal coordinate and phylogenetic analyses

For estimation of probable genetic organisation of studied rice accessions, STRUCTURE v2.3.4 based population structure analysis was carried out and results were further confirmed by comparing with principal coordinate (PCoA) and phylogenetic analyses. The highest value of ΔK was found at K = 3 (S1 Fig in S1 File), grouping all rice accessions into three distinct subpopulations (SPs), designated here as SP1 (red), SP2 (blue) and SP3 (green) (S2 Fig in S1 File). The PCoA (Fig 3) and UPGMA phylogenetic (Fig 4) analyses also supported population structure results. The SP1 and SP2 harboured 20 and 10 fine grain accessions. Likewise, SP3 also harboured 25 fine and coarse grain accessions. Unexpectedly, one Chinese-originated coarse grain genotype (UHL17078; serial # 51) also clustered with fine grain genotypes of SP2 (Figs 3 and 4), which could be due to more similar DNA banding patterns of this genotype with fine grain accessions than with coarse grain genotypes. In UPGMA phylogenetic tree, more than 72% (16) of the total (22) high grain Fe and Zn rice accessions (containing > 30 ppm) were clustered into SP1 and SP2, whereas less than 28% (6) grouped into SP3 (Fig 4). Collectively, these results strongly indicate that fine grain rice accessions contain relatively higher micronutrient contents as compared with coarse grain accessions.

Subpopulations are represented with red (SP1), blue (SP2) and green (SP3) node colours. The scale indicates dissimilarity value among rice accessions. Markers with different colours indicate high Fe and Zn (> 30 ppm; black circle), high Fe and medium Zn (Zn 10–30 ppm; red circle), medium Fe and Zn (10–30 ppm; dark yellow square), medium Fe and high Zn (purple triangle), low Fe and high Zn (Fe < 10 ppm, blue triangle) and low Fe and medium Zn accessions or vice versa. The tree was generated with MEGA7 software.

## Associations among markers, their alleles and grain Fe-Zn contents

For testing of genetic associations among 21 polymorphic markers and grain Fe-Zn contents, single-factor ANOVA based single marker analysis was accomplished. Statistically

**Table 1. Summary statistics of genetic diversity parameters among 24 SSR markers used in this study.**

| Marker | $N_A$ | $A_R$ | $M_{AF}$ | $G_D$ | $H_O$ | PIC | f |
|---|---|---|---|---|---|---|---|
| RM 319 | 2 | 133–142 | 0.8393 | 0.2813 | 0.0000 | 0.2610 | 1.0000 |
| RM 152 | 2 | 140–151 | 0.4821 | 0.5697 | 0.0179 | 0.4752 | 0.9692 |
| RM 444 | 3 | 155–240 | 0.6250 | 0.5403 | 0.0357 | 0.4843 | 0.9350 |
| RM 7414 | 1 | 94 | 1.0 | - | - | - | - |
| RM 5607 | 3 | 90–107 | 0.7143 | 0.4224 | 0.0179 | 0.3535 | 0.9585 |
| RM 211 | 5 | 140–190 | 0.6161 | 0.5778 | 0.0714 | 0.5444 | 0.8785 |
| RM 335 | 9 | 80–160 | 0.2679 | 0.8182 | 0.0536 | 0.7960 | 0.9357 |
| RM 273 | 2 | 200–207 | 0.8125 | 0.3047 | 0.0893 | 0.2583 | 0.7114 |
| RM 303 | 6 | 150–200 | 0.4732 | 0.6386 | 0.0179 | 0.5744 | 0.9725 |
| RM 153 | 1 | 201 | 1.0 | - | - | - | - |
| RM 430 | 4 | 140–173 | 0.5000 | 0.5965 | 0.1786 | 0.5179 | 0.7052 |
| RM 437 | 4 | 250–280 | 0.5714 | 0.5383 | 0.1071 | 0.4496 | 0.8042 |
| RM 31 | 6 | 130–160 | 0.5357 | 0.6634 | 0.1607 | 0.6336 | 0.7616 |
| RM 190 | 5 | 110–130 | 0.4286 | 0.7329 | 0.0357 | 0.6977 | 0.9521 |
| RM 402 | 3 | 110–140 | 0.5446 | 0.6041 | 0.0893 | 0.5409 | 0.8547 |
| RM 541 | 4 | 158–185 | 0.5357 | 0.6346 | 0.0000 | 0.5874 | 1.0000 |
| RM 501 | 4 | 160–200 | 0.3482 | 0.6905 | 0.1964 | 0.6279 | 0.7199 |
| RM 560 | 3 | 239–260 | 0.6071 | 0.5032 | 0.0000 | 0.4085 | 1.0000 |
| RM 234 | 3 | 130–156 | 0.5179 | 0.5466 | 0.0714 | 0.4466 | 0.8715 |
| RM 1357 | 1 | 126 | 1.0 | - | - | - | - |
| RM 331 | 3 | 150–176 | 0.5714 | 0.5810 | 0.0714 | 0.5176 | 0.8791 |
| RM 242 | 2 | 200–225 | 0.5536 | 0.5780 | 0.1250 | 0.5036 | 0.7872 |
| RM 439 | 3 | 250–300 | 0.9107 | 0.1677 | 0.0000 | 0.1630 | 1.0000 |
| RM 3412 | 6 | 200–245 | 0.2857 | 0.8042 | 0.0714 | 0.7765 | 0.9127 |
| Mean | 3.5 | - | 0.5956 | 0.5203 | 0.0588 | 0.4663 | 0.8889 |
| SD ± | 1.8 | - | 0.2051 | 0.2323 | 0.0577 | 0.2183 | 0.0967 |

$N_A$; the number of alleles, $A_R$; amplicon range, $M_{AF}$; major allele frequency, $G_D$; gene diversity, $H_O$; observed heterozygosity, PIC; polymorphism information content, f; inbreeding coefficient, SD; standard deviation.

significant genetic associations were observed among four polymorphic markers (RM 152, RM 234, RM 335 and RM 5607) and grain Zn contents (Table 2). Similarly, grain Fe contents showed significant association with RM 501 polymorphic marker. These five significant marker-trait associations (MTAs) accounted for 15.41–39.72% of the total observed phenotypic variance ($R^2$). We also compared different alleles of five significantly associated polymorphic markers with their respective Fe and Zn contents to identify high grain micronutrient linked alleles. For grain Zn contents, 142 bp (43 ppm), 130 bp (58 ppm) and 107 bp (29 ppm) alleles of RM 152, RM 234 and RM 5607 respectively, revealed significantly higher contents than all other respective alleles (Fig 5). Similarly, 80 bp (39 ppm) and 150 bp (41 ppm) alleles of RM 335 showed higher grain Zn contents as compared with all other alleles. Likewise, 200 bp (49 ppm) allele of RM 501 exhibited significantly higher Fe contents than three other respective alleles. Overall, significant MTAs indicate potential application of high micronutrients linked alleles for development of biofortified rice cultivars. In future, it would be interesting to validate these marker results using diverse and large germplasm sets and identify high grain Fe and Zn linked alleles for the advancement of marker-assisted biofortification breeding efforts.

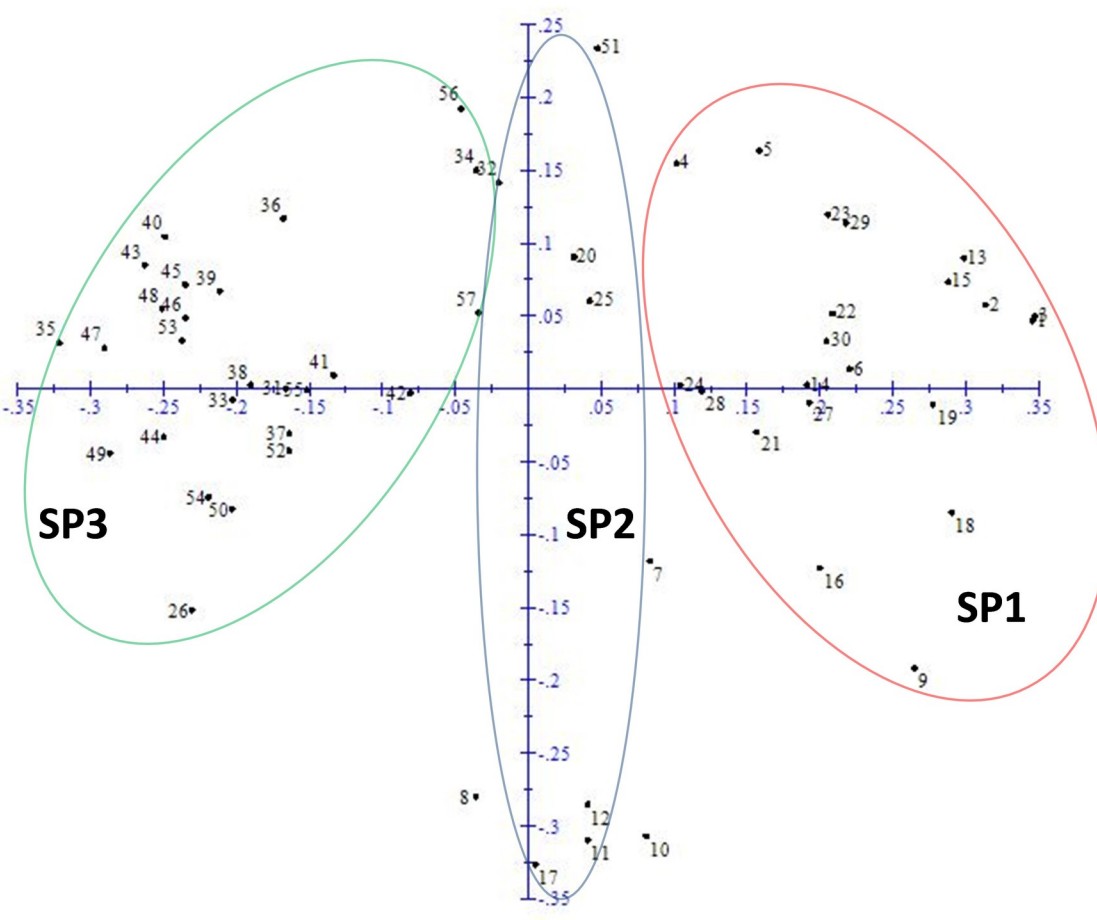

**Fig 3. Principal coordinate analysis of 56 rice accessions.**

## Discussion

Knowledge of genetic variation in grain Fe and Zn contents and genetic relationships between genotypes are crucial for their possible utilization in rice grain quality improvement. Like other crop plants, molecular markers have been proved to be most effective in evaluating genetic diversity, structure analysis and phylogenetic relatedness among rice accessions too [10, 36]. During recent years, SSR markers based genotyping has proven to be very useful for exploring a variety of surreptitious information in plants, ranging from domestication traits to crop improvement through marker-assisted breeding [37–40]. Although, individual reports on micronutrient density and SSR based genetic diversity in rice have been recently published [10, 12, 13, 36, 41], however, to date, a combined study of grain Fe and Zn contents and trait linked SSR markers based genetic diversity is obscure.

In this study, we assessed genetic diversity in the representative subset of a large collection of local and exotic rice germplasm for grain Fe and Zn contents using trait linked SSR markers. Grain mineral and protein contents change with a change in edaphic factors and genotype x environment interactions [42]. Therefore, pooled grain Fe and Zn data of two years is more reliable than single-year data. Significant genetic variation was observed among 56 studied rice accessions for Fe and Zn contents in brown rice (Fig 1A). Sing and Sing [13] observed wide

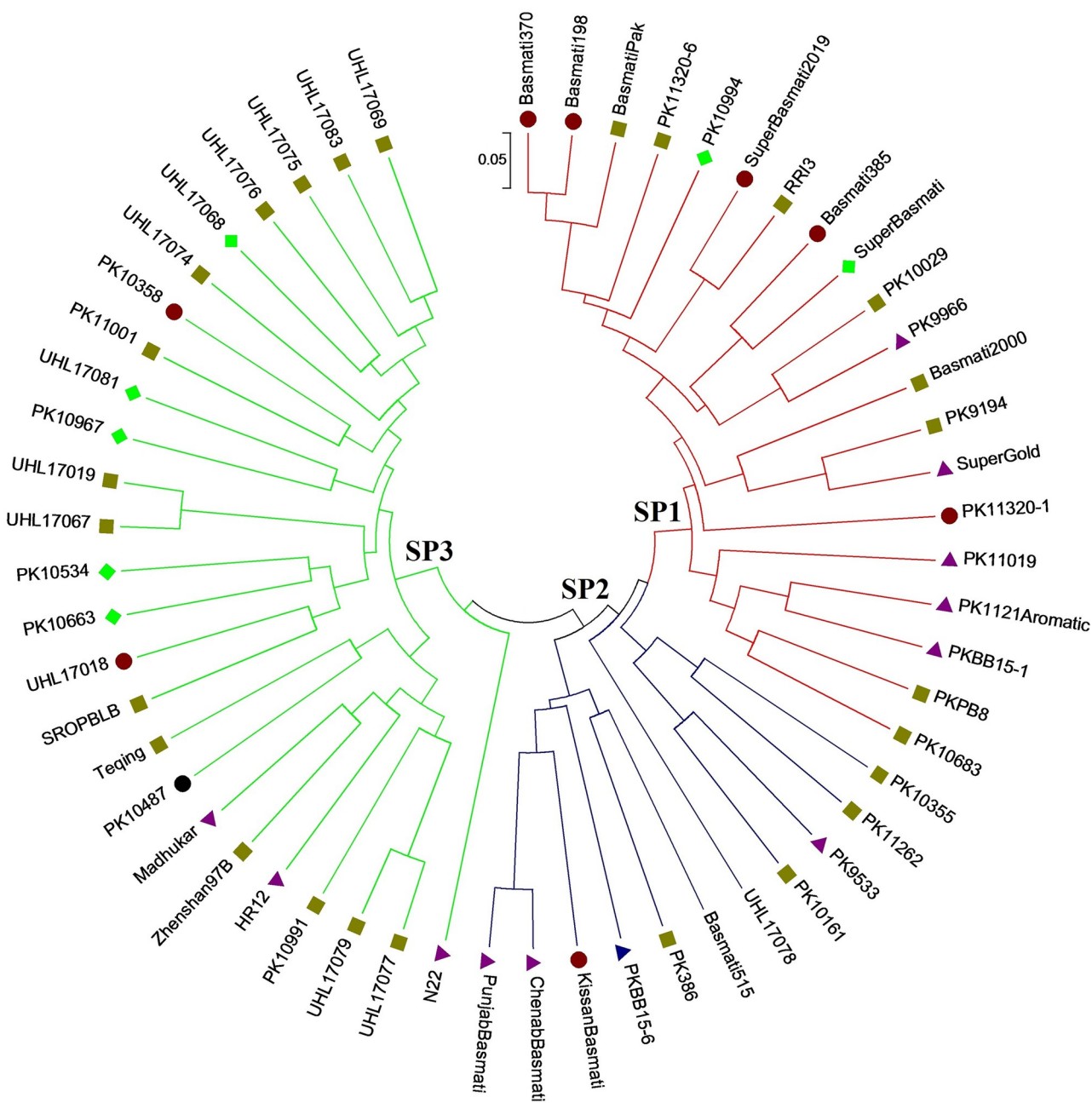

**Fig 4. UPGMA phylogenetic tree showing relationships among 56 rice accessions.**

genetic diversity in dehusked rice seeds for Fe (11.42–252.62 µg/g) and Zn (17.98–75.8 µg/g) contents. Similarly, Bollinedi et al. [36] also reported significant genetic variation in Fe and Zn contents of brown and milled rice samples. These reports support results presented in this study. The Fe and Zn contents in unpolished grains of fine/aromatic rice accessions were relatively higher than coarse grain accessions (Fig 1B). Gregorio et al. [43] and Verma and Srivastav [44] also reported higher Fe and Zn contents in aromatic rice cultivars than non-aromatic cultivars. Significant positive correlation was present between Fe and Zn contents (Fig 1C). Several other studies have also reported a positive association between these two mineral

Table 2. Summary statistics of MTA analysis.

| Marker | p-value | | $R^2$ (%) |
|---|---|---|---|
| | Fe | Zn | |
| RM 319 | 0.2857 | 0.2551 | - |
| RM 152 | 0.4061 | **0.0454**[*] | 22.71 |
| RM 444 | 0.6592 | 0.9847 | - |
| RM 5607 | 0.4961 | **0.0129**[*] | 15.41 |
| RM 211 | 0.5035 | 0.1494 | - |
| RM 335 | 0.5435 | **0.0132**[*] | 39.72 |
| RM 273 | 0.6069 | 0.2841 | - |
| RM 303 | 0.6041 | 0.5620 | - |
| RM 430 | 0.4049 | 0.0705 | - |
| RM 437 | 0.2926 | 0.3223 | - |
| RM 31 | 0.6692 | 0.8720 | - |
| RM 190 | 0.4660 | 0.9636 | - |
| RM 402 | 0.2545 | 0.1092 | - |
| RM 541 | 0.6139 | 0.8183 | - |
| RM 501 | **0.0093**[*] | 0.4732 | 33.52 |
| RM 560 | 0.0548 | 0.3315 | - |
| RM 234 | 0.2323 | **0.0056**[*] | 21.70 |
| RM 331 | 0.8934 | 0.7243 | - |
| RM 242 | 0.2457 | 0.2292 | - |
| RM 439 | 0.0593 | 0.9578 | - |
| RM 3412 | 0.1024 | 0.5883 | - |

contents [10, 36, 45, 46], which could be explained by co-localization of Fe and Zn responsive QTLs/genes [29] and possible pleotropic effects of metal transporter proteins encoding genes. Overall, these results indicate the prevalence of wide genetic variability in studied rice accessions for grain Fe and Zn contents and possibility of their simultaneous improvement. However, mineral densities in polished or white rice need to be assessed before utilizing these accessions in rice biofortification breeding programs, as loss of significant quantities of both micronutrients have been reported during the polishing process [12, 36, 47].

The studied rice accessions contain considerable genetic diversity as revealed by grain Fe and Zn contents linked SSR markers (Table 1). More than 87% of the total markers used in this study were found polymorphic. The 3.5 average numbers of alleles per locus were comparable with 2.44 and 3.96 previously reported by Pradhan et al. [10] and Upadhyay et al. [48], respectively. However, it was lower than that reported by Singh et al. [13] (6.77 alleles per locus). Yadav et al. [49] reported higher genetic diversity and PIC values in some Indian rice germplasm using trait linked SSR markers. In this study, nearly 70% (17) markers had greater than 0.4085 gene diversity and PIC values, strongly indicating that majority of the markers used in current research are moderate to highly informative for evaluating genetic diversity [50]. These results are further supported by higher inbreeding coefficient values (range; 0.7052–1.0000) (Table 1).

Genetic relationships among fine and coarse grain accessions were assessed through population structure, principal coordinate (PCoA) and phylogenetic analyses. Structure analysis classified studied 56 rice accessions into three (highest ΔK value at K = 3) distinct sub-populations (red, SP1; blue, SP2; green, SP3) (S1 and S2 Figs in S1 File). Similar clustering patterns were revealed by PCoA (Fig 3) and phylogenetic (Fig 4) analyses. Pradhan et al. [10] also

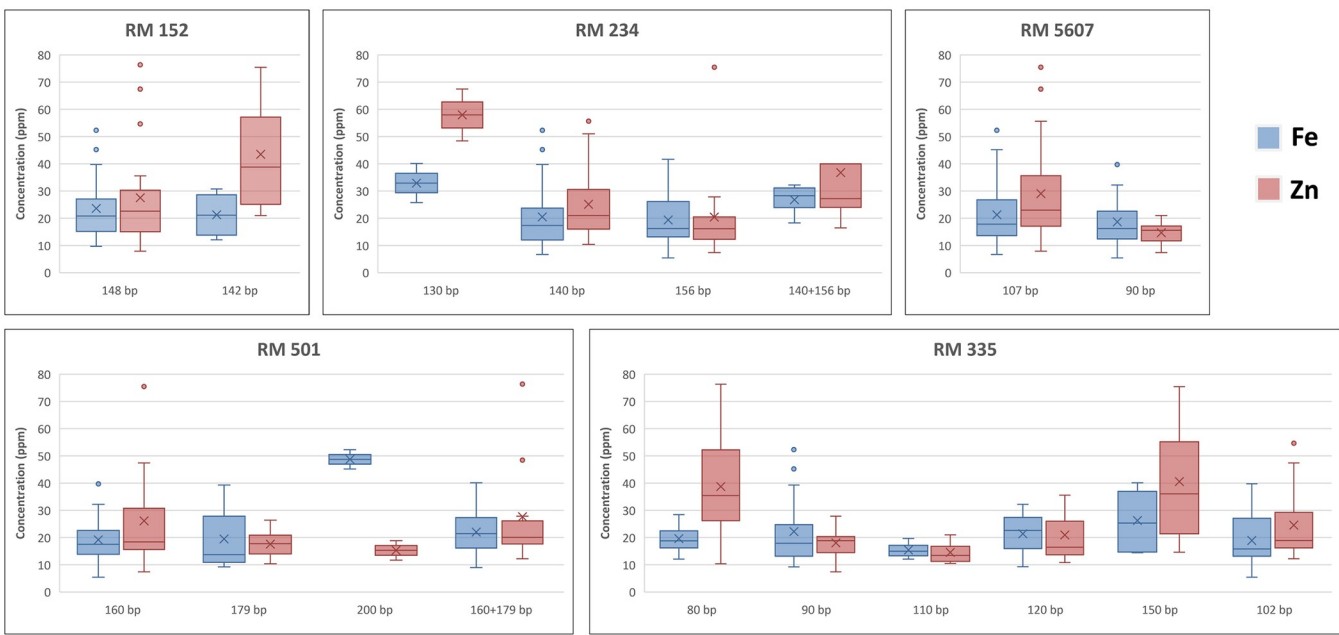

**Fig 5. Comparison of significant MTA alleles with grain Fe and Zn contents.**

clustered a panel population of 102 diverse rice accessions into three subpopulations (SPs). Based upon marker data, the SP1 and SP2 are seems to be fine or aromatic grain specific, except for a Chinese originated coarse grain accession UHL17078. Whereas, SP3 harboured admixture of both fine and coarse grain accessions. These results suggest that trait linked markers used in this study might be also good for genetic differentiation between fine and coarse grain genotypes. Moreover, the majority of the high grain Fe and Zn accessions (containing > 30 ppm) were grouped into SP1 and SP2 (Fig 4), strongly supporting "micronutrients enriched aromatic grain" hypothesis. Previous reports also support our new hypothesis [18, 43, 44]. Although, studies involving $F_2$-derived populations demonstrated no pleiotropic effect of aroma on grain Fe and Zn contents, however, aroma trait may be used for screening of high grain Fe and Zn contents rice germplasm [18]. In future, it would be interesting to investigate comparative micronutrient densities in aromatic and non-aromatic rice accessions along with complex underlying genetic and molecular mechanisms.

Marker-trait associations (MTAs) offer clues about the existence of trait linked QTLs/genes in diverse genetic backgrounds. In marker-assisted breeding programs, a strong MTA is preferred over weak MTA to meritoriously exploit that particular marker for trait improvement [51]. In this research, single-marker analysis exhibited significant ($P < 0.05$) associations among grain Zn contents and RM 152, RM 234, RM 335 and RM 5607 (Table 2). Whereas, only single marker RM 501 could be significantly associated with grain Fe contents. Their higher phenotypic variance values indicate that they control a considerable amount of genetic variation in grain mineral contents and could be reliable genetic markers for further improvement of Fe and Zn contents in rice grains. All these markers have been variably reported to be linked with grain micronutrients contents [10, 29, 52–55].

The co-dominance nature of SSR markers makes them ideal candidates for segregation analysis in large population sets [24, 41, 56]. Different alleles of the same polymorphic marker show variable effects on genetic variation of linked traits [57]. This fact prompted us to compare grain Fe and Zn contents with different alleles of significantly associated polymorphic

markers. For grain Zn contents, single alleles of RM 152 (142 bp), RM 234 (130 bp) and RM 5607 (107 bp) and two duplicated alleles of RM 335 (80 bp and 150 bp) demonstrated relatively higher contents than their all other respective alleles (Fig 5). Similarly, significantly higher grain Fe contents were observed in 200 bp allele containing rice accessions, which were genotyped with RM 501. Our results suggest that these alleles could be used for screening of high grain Fe and Zn containing rice germplasm and marker-assisted development of biofortified rice cultivars. To the best of our knowledge, this is the first report on comparison of SSR marker alleles with grain micronutrient contents in rice. Therefore, these results should be validated using large and diverse germplasm sets before practical application in marker-assisted biofortification breeding in rice.

## Conclusions

Rice is one of the major crops for sustainable food and nutritional security. However, trace micronutrients quantities in grains aggregate malnutrition in rice-eating poor populations. Plant breeding based biofortification of edible plant parts can help in eradicating micronutrient deficiencies. In this study, we found wide genetic variation in grain Fe and Zn contents of genetically diverse rice accessions representing a large collection of local and exotic rice germplasm. Significant positive correlation was observed between Fe and Zn contents in unpolished grains, indicating possibility for simultaneous improvement of both mineral elements. Genotyping results revealed that markers used in current research were not only informative for genetic diversity study but can also be employed for genetic differentiation between fine and coarse grain rice accessions. Moreover, population structure, PCoA and phylogenetic analyses clustered studied rice accessions into fine and fine/coarse grain admixture subpopulations. Furthermore, significant marker-trait associations and high grain Fe and Zn contents linked marker alleles were also recognized, which could be exploited through marker-assisted breeding for development of biofortified rice cultivars. However, these results require validation using large and diverse germplasm sets before successful application in marker-assisted biofortification breeding.

## Supporting information

**S1 File.**
(RAR)

## Author Contributions

**Conceptualization:** Qasim Raza, Muhammad Sabar.

**Formal analysis:** Qasim Raza, Awais Riaz.

**Funding acquisition:** Muhammad Sabar.

**Methodology:** Qasim Raza, Awais Riaz, Hira Saher, Ayesha Bibi, Mohsin Ali Raza.

**Project administration:** Syed Sultan Ali, Muhammad Sabar.

**Resources:** Mohsin Ali Raza, Muhammad Sabar.

**Supervision:** Syed Sultan Ali, Muhammad Sabar.

**Writing – original draft:** Qasim Raza.

**Writing – review & editing:** Awais Riaz, Syed Sultan Ali, Muhammad Sabar.

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
