## [Decision Letter · Decision Letter 0]

24 Jul 2020

PONE-D-20-17969

Grain Fe and Zn contents linked SSR markers based genetic diversity reveal perspective for marker assisted biofortification breeding in rice

PLOS ONE

Dear Dr. Raza,

Thank you for submitting your manuscript to PLOS ONE. After careful consideration, we feel that it has merit but does not fully meet PLOS ONE’s publication criteria as it currently stands. Therefore, we invite you to submit a revised version of the manuscript that addresses the points raised during the review process.

The manuscript has scope for publication but reviewer#2 raised some valid questions about the sample size used during the present study. You are therefore requested to revise the manuscript as per suggestions made by both the reviewers and resubmit the manuscript to PLOS ONE

We look forward to receiving your revised manuscript.

Kind regards,

Reyazul Rouf Mir, PhD

Academic Editor

PLOS ONE

Journal Requirements:

2.

PLOS ONE now requires that authors provide the original uncropped and unadjusted images underlying all blot or gel results reported in a submission’s figures or Supporting Information files. This policy and the journal’s other requirements for blot/gel reporting and figure preparation are described in detail at https://journals.plos.org/plosone/s/figures#loc-blot-and-gel-reporting-requirements and https://journals.plos.org/plosone/s/figures#loc-preparing-figures-from-image-files. When you submit your revised manuscript, please ensure that your figures adhere fully to these guidelines and provide the original underlying images for all blot or gel data reported in your submission. See the following link for instructions on providing the original image data: https://journals.plos.org/plosone/s/figures#loc-original-images-for-blots-and-gels.

Additional Editor Comments (if provided):

The manuscript was reviewed by two reviewers. While both reviewers DISCOVERED scope in this work and made some useful suggestions for the improvement of this manuscript, I am PARTICULARLY worried for reviewer # 2 comments where the reviewer suggested less number of genotypes have been used in this study. I am wondering whether authors can use additional genotypes in the study or can they explain in detail if the genotypes used by them in the present study are diverse and represent genetic diversity available in large collection of germplasm. I therefore advise authors to revise the manuscript carefully and address all comments by both the reviewers before re-submission to PLOS ONE.

Reviewers' comments:

Reviewer's Responses to Questions

**Comments to the Author**

1. Is the manuscript technically sound, and do the data support the conclusions?

Reviewer #1: Yes

Reviewer #2: Partly

2. Has the statistical analysis been performed appropriately and rigorously? 

Reviewer #1: Yes

Reviewer #2: Yes

3. Have the authors made all data underlying the findings in their manuscript fully available?

Reviewer #1: Yes

Reviewer #2: Yes

4. Is the manuscript presented in an intelligible fashion and written in standard English?

Reviewer #1: Yes

Reviewer #2: Yes

5. Review Comments to the Author

Reviewer #1: Title of the manuscript is too long and there is the scope to cut it short

The manuscript has some typing errors and grammatical mistakes. Kindly ensure the correction.

The authors may clarify whether the markers used are sufficient to evaluate the genetic/genomic diversity. Moreover, are these markers distributed throughout the genome or they are located only at certain regions.

In discussion section, simply by saying our findings are in accordance with that of previous workers is not sufficient. Some elaboration of the work of previous researchers need to be carried out.

Conclusion is the weakest section of the manuscript and need to be rewritten

Reviewer #2: In the research article entitled “Grain Fe and Zn contents linked SSR markers based genetic diversity reveal perspective for marker assisted bio-fortification breeding in rice”, authors have studied genetic diversity of 56 indica rice genotypes (both fine and coarse grain type) using 24 previously reported SSR markers linked to Fe and Zn content of rice, estimated Fe & Zn content in two cropping seasons.

I present here some of my observations

Abstract needs revision and clarity.

Did authors use known standard check for Fe & Zn estimation, what is the status of polished rice Fe and Zn content of 56 genotypes?

I think only 56 genotypes for diversity, population structure and phylogenetic analysis is very less number

Line No: 62 resent to recent

Line No: 73,124 indica to indica

Line No 50: The sentence “Rice genotypes of South Asia possess some unique

characteristics which are of great interest to the modern rice breeders” It is a very general statement and authors have listed abiotic stresses like heat and drought tolerance as unique characters. It requires more clarity or detailed explanation.

Line No 56:

Like other crop plants, several studies have been conducted in rice to evaluate genetic diversity using molecular markers [10, 13, 22, 23]. Specific trait - Fe and Zn genetic diversity is missed in the above sentence.

Language has scope for further improvement

Is there any earlier reports available for coarse grains having less Fe & Zn than fine grains may be presented.

6. PLOS authors have the option to publish the peer review history of their article (what does this mean?). If published, this will include your full peer review and any attached files.

Reviewer #1: **Yes: **Dr Showkat A Waza, Assistant Professor/ Scientist

Reviewer #2: No

---

## [Author Response · Author response to Decision Letter 0]

4 Aug 2020

Editor Comments: The manuscript was reviewed by two reviewers. While both reviewers DISCOVERED scope in this work and made some useful suggestions for the improvement of this manuscript, I am PARTICULARLY worried for reviewer # 2 comments where the reviewer suggested less number of genotypes have been used in this study. I am wondering whether authors can use additional genotypes in the study or can they explain in detail if the genotypes used by them in the present study are diverse and represent genetic diversity available in large collection of germplasm. I therefore advise authors to revise the manuscript carefully and address all comments by both the reviewers before re-submission to PLOS ONE.

Response: Thank you and both reviewers for detailed comments and suggestions. The genotypes used in this study were derived from two major rice groups (Indica and Japonica) and originated from three major rice producing countries (China, India and Pakistan). These included landraces, farmer field improved cultivars, advance uniform lines in release pipeline, recombinant/near-isogenic/backcross inbred lines, fine grain, coarse grain, aromatic, non-aromatic and few sequenced genome accessions. Although, the number of genotypes seems to be less, however these are genetically diverse and representative of a large collection of local and exotic rice germplasm. That’s why we think their numbers are sufficient for carrying out genetic diversity study.

Reviewer 1

Comment 1: Title of the manuscript is too long and there is the scope to cut it short.

Response: Title has been changed to “Grain Fe and Zn contents linked SSR markers based genetic diversity in rice”. Line 1–2 in final manuscript.

Comment 2: The manuscript has some typing errors and grammatical mistakes. Kindly ensure the correction.

Response: Thanks for highlighting the issues. Typing errors and grammatical mistakes were corrected in revised manuscript.

Comment 3: The authors may clarify whether the markers used are sufficient to evaluate the genetic/genomic diversity. Moreover, are these markers distributed throughout the genome or they are located only at certain regions.

Response: Markers used in current research were taken from our previous study (Raza et al. 2019, Plant Science), in which we performed a comprehensive meta-analysis on all grain Fe and Zn contents associated QTLs (reported upto May, 2019) and identified meta-QTLs and candidate genes for breeding of biofortified rice. Although, we selected moderate number of randomly distributed markers across the genome, nevertheless these were trait-specific and demonstrated higher gene diversity and polymorphism information content values, strongly indicating that these were moderate to highly informative for evaluating the genetic diversity.

Comment 4: In discussion section, simply by saying our findings are in accordance with that of previous workers is not sufficient. Some elaboration of the work of previous researchers needs to be carried out.

Response: In revised manuscript, we have incorporated suggested changes which provide elaboration of previous related works. Line 241–244 & 249–251 in final manuscript.

Comment 5: Conclusion is the weakest section of the manuscript and need to be rewritten.

Response: This section was improved in the revised manuscript and conclusions were drawn appropriately based on the data presented. Line 311–326 in final manuscript.

Reviewer 2

Comment 1: Abstract needs revision and clarity.

Response: Abstract has been revised for more clarity. Line 15–32 in final manuscript.

Comment 2: Did authors use known standard check for Fe & Zn estimation, what is the status of polished rice Fe and Zn content of 56 genotypes?

Response: No, sorry we could not afford the price of lab grade standard check material for Fe & Zn estimation. However, for quality control we repeated the analyses thrice separately for Fe & Zn over the two seasons and considered mean values for final analysis. Additionally, few landraces (Basmati 370), Basmati 198, Basmati Pak) and commercial check varieties (Super Basmati, Basmati 515, PK 1121 Aromatic) were also included in studied material for quality comparison.

Results of polished rice Fe and Zn contents are not available; however it is reasonable to assume that genotypes with higher Fe and Zn contents in brown rice might also have adequate contents in polished rice.

Comment 3: I think only 56 genotypes for diversity, population structure and phylogenetic analysis is very less number.

Response: The 56 genotypes used in this study were derived from two major rice groups (Indica and Japonica) and originated from three major rice producing countries (China, India and Pakistan). These included landraces (Basmati 370, Basmati Pak, Basmati 198), farmer field improved cultivars (Basmati 515, Kissan Basmati, Super Basmati 2019), advance uniform lines (PK10683, PK9194, PK10029), recombinant inbred lines (PK10355, PK9533, PK10161), near-isogenic/backcross inbred lines (PKBB15-1, PKBB15-6, PKBB15-116), fine grain (Basmati 385, Super Basmati, PKPB8), coarse grain (SR OP BLB, Madhukar, Teqing), aromatic (Basmati 2000, PK 1121 Aromatic, RRI 3), non-aromatic (PK 386, UHL17019, UHL17067) and few sequenced genome accessions (Basmati 370, Zhenshan 97B, N22). Although, the number of genotypes seems to be less, however these are genetically diverse and representative of a large collection of local and exotic rice germplasm, which we think are sufficient for carrying out genetic diversity, population structure and phylogenetic analyses.

Comment 4: Line No: 62 resent to recent.

Response: Spelling mistake corrected in revised manuscript. Line 72 in final manuscript.

Comment 5: Line No: 73,124 indica to indica.

Response: Suggested changes have been incorporated in revised manuscript. Line 82 & 134 in final manuscript.

Comment 6: Line No 50: The sentence “Rice genotypes of South Asia possess some unique characteristics which are of great interest to the modern rice breeders” It is a very general statement and authors have listed abiotic stresses like heat and drought tolerance as unique characters. It requires more clarity or detailed explanation.

Response: For clarity, the sentence has been revised and now it states “Generally, due to great diversity in climatic and edaphic factors, rice genotypes of South Asia possess some unique characteristics which are of great interest to the modern rice breeders”. Line 58–60 in final manuscript.

Comment 7: Line No 56: Like other crop plants, several studies have been conducted in rice to evaluate genetic diversity using molecular markers [10, 13, 22, 23]. Specific trait - Fe and Zn genetic diversity is missed in the above sentence.

Response: Trait-specific genetic diversity information has been added and the revised sentence states “Like other crop plants, several studies have been conducted in rice to evaluate genetic diversity in grain Fe & Zn contents, drought tolerant & susceptible genotypes and local & exotic germplasm using molecular markers [10, 13, 22, 23]”. Line 65–67 in final manuscript.

Comment 8: Language has scope for further improvement.

Response: We have tried our best to improve the language in revised manuscript.

Comment 9: Is there any earlier reports available for coarse grains having less Fe & Zn than fine grains may be presented.

Response: Yes, Gregorio et al. (1999) and Verma and Srivastav (2017) have previously reported higher Fe and Zn contents in aromatic rice cultivars than non-aromatic cultivars. As majority of the aromatic/basmati cultivars are in fine grain background, it is reasonable to expect that aromatic/fine grain cultivars possess higher Fe & Zn contents than non-aromatic/coarse grain cultivars. Line 246–248 in final manuscript.

---

## [Decision Letter · Decision Letter 1]

14 Sep 2020

Grain Fe and Zn contents linked SSR markers based genetic diversity in rice

PONE-D-20-17969R1

Dear Dr. Raza,

We’re pleased to inform you that your manuscript has been judged scientifically suitable for publication and will be formally accepted for publication once it meets all outstanding technical requirements.

Kind regards,

Reyazul Rouf Mir, PhD

Academic Editor

PLOS ONE

Additional Editor Comments (optional):

Reviewers' comments:

Reviewer's Responses to Questions

**Comments to the Author**

1. If the authors have adequately addressed your comments raised in a previous round of review and you feel that this manuscript is now acceptable for publication, you may indicate that here to bypass the “Comments to the Author” section, enter your conflict of interest statement in the “Confidential to Editor” section, and submit your "Accept" recommendation.

Reviewer #1: All comments have been addressed

Reviewer #2: All comments have been addressed

2. Is the manuscript technically sound, and do the data support the conclusions?

Reviewer #1: Yes

Reviewer #2: Partly

3. Has the statistical analysis been performed appropriately and rigorously? 

Reviewer #1: Yes

Reviewer #2: Yes

4. Have the authors made all data underlying the findings in their manuscript fully available?

Reviewer #1: Yes

Reviewer #2: Yes

5. Is the manuscript presented in an intelligible fashion and written in standard English?

Reviewer #1: Yes

Reviewer #2: Yes

6. Review Comments to the Author

Reviewer #1: The comments raised have been addressed by authors. The manuscript is now in a good shape and should be accepted for publication.

Reviewer #2: In the presented study, though less number of genotypes have been utilized, seems to be genetically diverse origin,hence may be considered for publication

7. PLOS authors have the option to publish the peer review history of their article (what does this mean?). If published, this will include your full peer review and any attached files.

Reviewer #2: No

---

## [Editor Report · Acceptance letter]

16 Sep 2020

PONE-D-20-17969R1

Grain Fe and Zn contents linked SSR markers based genetic diversity in rice

Dear Dr. Raza:

I'm pleased to inform you that your manuscript has been deemed suitable for publication in PLOS ONE. Congratulations! Your manuscript is now with our production department.

Kind regards,

on behalf of

Dr. Reyazul Rouf Mir 

Academic Editor

PLOS ONE